# Role of Combined ^68^Ga DOTA-Peptides and ^18^F FDG PET/CT in the Evaluation of Gastroenteropancreatic Neuroendocrine Neoplasms

**DOI:** 10.3390/diagnostics12020280

**Published:** 2022-01-22

**Authors:** Chalermrat Kaewput, Sobhan Vinjamuri

**Affiliations:** 1Division of Nuclear Medicine, Department of Radiology, Faculty of Medicine Siriraj Hospital, Mahidol University, Bangkok 10700, Thailand; 2Department of Nuclear Medicine, Royal Liverpool University Hospital, Liverpool L7 8XP, UK; sobhan.vinjamuri@gmail.com

**Keywords:** GEP-NENs, FDG PET, SSTR PET, neuroendocrine tumors

## Abstract

This review article summarizes the role of combined ^68^Ga DOTA-peptides and ^18^F-fluorodeoxyglucose (FDG) positron emission tomography/computed tomography (PET/CT) in the evaluation of gastroenteropancreatic neuroendocrine neoplasms (GEP-NENs). Patients with GEP-NENs can initially present themselves to a gastroenterologist or endocrinologist rather than cancer specialist; hence, it is vital for a wider group of clinicians to be familiar with the range of tests available for the evaluation of these patients. The role of PET scanning by using ^68^Ga DOTA-peptides has a high sensitivity in the diagnosis of GEP-NENs and to guide patient selection for treatment with somatostatin analogues (SSA) and/or peptide receptor radionuclide therapy (PRRT). The loss of somatostatin receptor (SSTR) expression was found to be associated with an increased glucose metabolism in cells. However, the routine use of SSTR targeted radiotracers in combination with ^18^F-FDG to evaluate glucose utilization in GEP-NENs is still debatable. In our opinion, in patients with NENs, ^18^F-FDG PET should be performed in the case of a negative or slightly positive ^68^Ga DOTA-peptides PET scan for assessing the dedifferentiation status, to guide correct therapeutic strategy and to evaluate the prognosis. The approach of combined receptor and metabolic imaging can improve diagnostic accuracy, especially considering the heterogeneity of these lesions. Therefore, ^68^Ga DOTA-peptides and ^18^F-FDG PET should be considered complementary in patients with GEP-NENs.

## 1. Introduction

Neuroendocrine neoplasms (NENs) are epithelial neoplasms that arise from cells of the endocrine and nervous systems, with distinctive differentiation, and could occur in the various organ but more frequently from the gastrointestinal tract, followed by the lung. In the past, these tumors were thought to be extremely rare, but many recent studies have shown an increasing incidence and prevalence, and the incidence of GEP-NENs has increased more than six-fold between 1997 and 2012 [1,2,3]. This increase is partly due to better diagnostic methods and greater awareness of the disease.

Most NENs express somatostatin receptors (SSTR), which could be used as a target for radionuclides imaging and treatment. Somatostatin (SST) is a cyclic peptide hormone, also known as somatotropin release inhibiting factor (SRIF). It was originally found to be a growth hormone inhibitor, but it is currently known to be involved in inhibiting various metabolic processes related to neurotransmitters and endocrine secretions, as well as controlling exocrine secretions [4]. SSTRs are widely expressed throughout the body, with five subtypes reported (SSTR 1–5). For most NENs, SSTR2 is the most predominantly expressed receptor subtype [5]. Innovations in somatostatin receptor-based technology have pushed forward molecular diagnostics and treatment of NENs, markedly improving the quality of life of many patients.

GEP-NENs are NENs occurring at various anatomical sites of the gastrointestinal tract and pancreas, and they comprise a heterogeneous group of epithelial neoplasms with a wide range of aggressiveness. According to the WHO 2019 classification system, GEP-NENs are classified according to the Ki-67 index and mitotic count. An important feature of the new classification is a difference between differentiated neuroendocrine tumors (NETs) and poorly differentiated neuroendocrine carcinomas (NECs), since both share common expression of neuroendocrine markers. These dichotomous morphological subdivisions into NETs and NECs are also differentiated by genetic evidence at specific anatomic sites, clinical, epidemiologic, histological differences, and prognosis. NETs are classified as G1, G2, or G3 in many organ systems, and they rely on the mitotic count and/or Ki-67 labeling index, and/or the presence of necrosis. The distinction of well-differentiated NET G3 from NEC, previously applicable only for Pan-NENs, is now used for gastrointestinal NENs in the revised WHO classification for GEP-NENs, as described in Table 1 [6].

NETs arising at various anatomical areas of the gastrointestinal tract display tumor identities that vary in their physiology and clinical presentation (Table 2). Specific staining for peptide hormones could be used to confirm the origin of clinical symptomatology, but there is no definite agreement between immunohistochemistry (IHC) and symptoms, since bioactive compounds can be synthesized without secretion or non-functioning NENs (NF-NENs).

There are several prognostic predictors of NENs, including pathological factors (cytology and Ki-67 index) and biological factors (chromogranin A) [8]. In general, NENs have a wide range of cell differentiation. The presence of receptors at the cell surface appears to depend on the differentiation of tumor cell, with well-differentiated tumors showing a higher affinity for somatostatin [9].

Imaging acts as a major role in evaluating NENs. These include detecting, staging, evaluating the response, and prognosis [10]. Relying on the site of tumor origin, NENs might show site-specific symptoms. Due to these variations, NENs patients can present with various stage, and over 50% have metastases at the time of diagnosis. Functioning NENs tend to present earlier than nonfunctioning NENs. Therefore, the imaging modalities need to be very different, and they need to be tailor-made for each patient depending on their clinical presentation.

### 1.1. Conventional Imaging

Conventional imaging (CI) methods, including contrast-enhanced CT scans and MRI, are usually used as the primary imaging methods for patients presenting with signs or symptoms suggestive of GEP-NENs, both for detecting the site of primary tumor and possible metastatic sites. These modalities are most useful for staging and surgical planning, due to the great anatomic detail of the tumors and their surrounding structures. Most GEP-NENs are shown as contrast-enhancing lesions during the arterial phase on CT or MRI. MRI is an ideal imaging modality for differentiating between benign and malignant hepatic lesions. MRI seems to have higher sensitivity than CT in the diagnosis of liver and bone-marrow metastases, but both types of imaging techniques may miss small lymph node metastases. The sensitivity of techniques can also be variable for either recurrent or metastatic malignancy. This is due to the fact that these techniques primarily detect change in anatomy rather than function, and, therefore, they are considered to have moderate specificity [11].

According to diagnostic and therapeutic guidelines for GEP-NENs, the sensitivity of CT imaging in the diagnosis of pancreatic tumors is approximately 70%, and specificity is approximately 90%. In addition, CT has high sensitivity and specificity in the detection of metastases. The sensitivity is 82% and the specificity is 92% in liver metastases, while the sensitivity is 60–70% and specificity is 87–100% for lymph nodes metastases [12,13]. CT is preferable for imaging the lungs as compared to MRI, because it has a better spatial resolution. It is considered as the most effective imaging method for the detection of small bowel NENs. The disadvantage of contrast-enhanced CT is its poor sensitivity to detect small lesions less than 1 cm and for bone metastases (sensitivity about 50%). Recent studies have reported that MRI has more than 80% sensitivity to detect hepatic metastases [14,15,16].

### 1.2. ^111^In-Pentetreotide

Functional imaging studies based on the expression of SSTRs play an important role in evaluating patients with suspected GEP-NENs and high density of SSTRs. CT and MRI allow for anatomical characterization, but they cannot provide accurate functional information and are often insufficient for the diagnosis of GEP-NENs if the lesions are small and cover various anatomical locations. Somatostatin receptor scintigraphy (SRS) can provide important functional data. Thus, SRS shows good accuracy for whole-body imaging and has been routinely used for the diagnosis and follow-up of GEP-NENs [17].

SRS using the somatostatin analog ^111^In-pentetreotide (^111^In-DTPA-D-Phe-conjugate of octreotide; Octreoscan^®^) was the first radiotracer agent to be approved for NENs in 1989 [18]. The radiotracer preferentially binds to SSTR subtypes 2, 3 and 5, especially 2, and has proven to be an essential diagnostic functional imaging for the diagnosis, staging and follow-up of NENs, both pulmonary and, in particular, GEP-NENs. The added value of a whole body assessment in one setting is particularly advantageous.

^111^In-Pentetreotide scans are routinely obtained at 4 and 24 h after the intravenous administration of the radiotracer, and image acquisition is ideally performed with a double-headed gamma camera equipped with a medium-energy collimator [18]. However, in spite of the advantage over CI, the principal limitation of using ^111^In-Pentetreotide is the low spatial resolution of the gamma camera that significantly affects the assessment of small lesions, particularly in organs with a high physiological uptake (such as the liver). The use of SPECT/CT imaging has markedly improved the accuracy of ^111^In-Pentetreotide, increasing its spatial resolution and the anatomic localization of pathologic sites of increased uptake due to CT co-registration, with decreased false positive results and its capacity to provide cross-sectional scintigraphic images [19,20,21,22,23].

^111^In-Pentetreotide, acquired as SPECT/CT, shows higher sensitivity and accuracy values than CI and can be still used for routine diagnostic imaging of non-functioning NEN-GEPs, with significant impact on treatment planning. The added value of SPECT/CT over CI was 35.6%. SPECT/CT corrected CI classification and patient management in 27.9% of cases, while it down-staged the disease better than CI in 9.6% of cases. However, the combined use of CI and functional imaging, using ^111^In-Pentetreotide SPECT/CT, still leaves a big gap in the diagnostic pathway [24]. Therefore, newer, higher-affinity SST analogs, labeled with radioisotopes with better resolution and dosimetry, such as ^68^Ga, which is a positron emitter, have been considered as promising SSTR imaging agents.

### 1.3. ^68^Ga-DOTA-Conjugated Peptide-Binding SSTR PET/CT

^68^Ga-DOTA-conjugated peptide-binding SSTR-based imaging techniques are acknowledged as a new category of radiopharmaceuticals that has recently been instructed into routine clinical practice, including DOTA-D Phe1-Tyr3-octreotide (DOTATOC), DOTA 1-Nal3-octreotide (DOTANOC), and DOTA-D Phe1-Tyr3-Thr8-octreotide (DOTATATE) [25]. Although all of these radiopeptides bind to SSTR2, each has various affinities for other SSTR subtypes. In addition to the greatest affinity to SSTR2, ^68^Ga-DOTA-NOC also reveals a good affinity for SSTR3 and SSTR5; ^68^Ga-DOTA-TOC also shows a high affinity for SSTR5, but less affinity compared to DOTA-NOC, whereas ^68^Ga DOTA-TATE predominantly binds to SSTR2 [26].

Table 3 lists the frequency of overexpression of different SSTR subtypes in different clinical presentations [27].

^68^Ga DOTA conjugated peptide-binding SSTR is generally used for primary tumor localization, including metastatic detection (staging) and evaluation of residual, recurrent, or progressive disease (restaging) [28,29] all the way through to the selection of patients for PRRT [30]. PRRT can successfully control symptoms due to hypersecretion of hormones and has also been displayed to improve OS in progressive or symptomatic NETs patients [31]. The mechanism of uptake is based on increased SSTR expression in neuroendocrine cells and can be detected on SSTR scanning [32]. The previous studies revealed statistically significant decreased tumor uptake of ^68^Ga-DOTA-peptides PET in poorly differentiated NETs, and its uptake might be related to aggressive behavior and might lead to poorer prognosis [33]. However, some cases in this study were low-grade NENs that showed low tumor uptake on ^68^Ga-DOTA-peptides PET/CT scans. It is possible that the FNA site may not fully reflect the true pathological grade of a patient with heterogeneity of cellular differentiation within the same tumor mass, and this may also reflect the potential ability of ^68^Ga-labeled SSTR PET/CT to map these cellular characteristics (Figure 1) [33,34].

The recommended indications for patients with NENs according to EANM Guidelines for ^68^Ga-DOTA-conjugated somatostatin PET/CT imaging [35] are as follows:Localized primary tumors and detect sites of metastasis (staging);Monitoring patients with known disease in the detection of residual, recurrent or progressive disease (re-staging);Determine SSTR status (patients with SSTR positive tumors are more likely to respond to targeted therapy with SSA);Select patients with metastasis for PRRT (^177^Lu or ^90^Y–DOTA-labelled peptides).

^68^Ga-DOTA-conjugated peptide-binding SSTR is significantly superior to ^111^In-Pentetreotide in both diagnostic accuracy and its therapeutic impact. Available evidence also supports the concept that ^68^Ga-DOTA-conjugated peptide-binding SSTR imaging often demonstrates tumor uptake in some patients with negative or equivocal ^111^In-Pentetreotide scans, and it identifies patients who may benefit from PRRT [36].

A prospective study of 131 GEP-NENs patients determined the superiority of ^68^Ga-DOTATATE PET/CT over ^111^In-pentetreotide SPECT/CT, CT, and/or MRI. ^68^Ga-DOTATATE PET/CT detected 95.1% of lesions, while CI detected 45.3% of lesions, and ^111^In-pentetreotide SPECT/CT detected only 30.9% of lesions with statistically significant differences between all imaging modalities. In the subgroup of patients with carcinoid symptoms but negative biochemical testing, ^68^Ga-DOTATATE PET/CT detected lesions in 65.2% of patients. Of note, 40% of the lesions detected by ^68^Ga-DOTATATE PET/CT were missed on CI or ^111^In-pentetreotide SPECT/CT. Additionally, based on the findings from ^68^Ga-DOTATATE PET/CT, there was a significant change in the patient management in approximately a third of the patients (32.8%) [37].

Binnebeek et al. revealed that significantly more lesions were detected on ^68^Ga-DOTATOC-PET/CT as compared to ^111^In-pentetreotide scans. The sensitivity for PET/CT was 99.9% (95% CI, 99.3–100.0), and for SPECT, it was 60.0% (95% CI, 48.5–70.2). The organ-by-organ analysis showed that the PET was most frequently visualized in the liver and skeleton. They concluded that ^68^Ga-DOTATOC-PET/CT is superior for detecting NENs compared to ^111^In-pentetreotide SPECT [38]. Additionally, treatment was changed in more than one-third of patients with SSTR PET/CT rather than ^111^In-pentetreotide scan [39].

### 1.4. ^18^F-FDG PET/CT

^18^F-FDG is used to evaluate glucose metabolism, and it is the most commonly used radiotracer in PET imaging. It is not currently routinely used for NENs imaging, but recent experience has suggested that it can provide complementary information. ^18^F-FDG PET/CT has a limited advantage in well-differentiated NETs, since most NENs frequently have near normal glucose metabolism [40]. ^18^F-FDG exploits the increased glycolytic activity of tumor cells, as glycolytic activity seems to be low in most NENs. In fact, it can differentiate between slowly proliferative and aggressive tumors [41,42]. Additionally, ^18^F-FDG PET/CT scans can allow prognostic prediction, including overall survival (OS) and progression free survival (PFS). Finally, it could potentially identify those who are not responding earlier, characterize early progression, and change therapy if required.

Assessment of glucose metabolism by ^18^F-FDG may be helpful in the diagnosis of high-risk aggressive disease associated with poor outcomes. ^18^F-FDG PET may still an essential modality in the management of NENs patients, due to its better prognostic value and greater sensitivity to identify the extent of disease, particularly in aggressive high-grade tumors [43] Bahri H et al. revealed that the median PFS and OS are significantly greater in patients with a negative ^18^F-FDG PET finding (OS nearly 120 months) than in patients with a positive ^18^F-FDG PET finding (only 15 months) [44]. Ezziddin et al. [45] retrospectively reviewed data from 89 patients with metastatic GEP-NENs and classified three different prognostic groups according to FDG uptake: mG1, tumor-to-liver ratio of maximum SUV ≤ 1.0; mG2, 1.0–2.3; and mG3, >2.3. These groups were correlated with significantly different OS (not more than 114 vs. 55 vs. 13 months, respectively). As mentioned, ^18^F-FDG PET must not only be used in staging but also for grading whole-body imaging modality; positive ^18^F-FDG PET lesions have a significant correlation with prognosis, irrespective of SSTR expression.

The recommended indications for ^18^F-FDG PET/CT in patients with NENs according to EANM Guideline for ^68^Ga-DOTA-conjugated somatostatin PET/CT imaging [35] are as follows:Localization of NECs and high-grade poorly differentiated NETs with aggressive behavior;Prediction of prognosis;Localization of synchrone/metachrone non-NEN malignancy.

### 1.5. Diagnostic Value of Combined ^68^Ga SSTR and ^18^F-FDG PET in GEP-NENs

The tumor grade indicates the tumor’s biological aggressiveness, which is recognized as a strong predictor for prognosis in many tumors. The NETs grading system is based on proliferation rate, which is determined by the percentage of tumor cells immunolabeled for the Ki-67 antigen positive (Ki-67 index) or the number of mitoses per 10 high-power microscopic fields or 2 mm^2^ (mitotic rate) [46]. Patients with high-grade tumors have high turnover with highly aggressive features and may predict rapid progression, while lower-grade patients may be able to live with a stable disease for a long time. It is essential to differentiate between rapidly progressing tumors and relatively stable tumors, because the treatment approaches for aggressive tumors might have long-term toxicity and only modest efficacy.

Due to the range of SSTR expressions between well-differentiated and poorly differentiated GEP-NENs, ^18^F-FDG PET/CT and ^68^Ga-DOTA-peptide PET/CT can have a complementary role in the imaging of NENs. Well-differentiated tumors tend to have greater ^68^Ga-DOTA-peptide avidity and lower FDG avidity (Figure 2). Conversely, poorly differentiated tumors tend to have lower ^68^Ga-DOTA-peptide avidity and greater FDG avidity [32,47]. These characteristics are referred to as the “flip-flop” phenomenon that is caused by greater SSTR expression and decreased glycolytic activity in well-differentiated tumors, and conversely decreased SSTR expression and greater glycolytic activity in poorly differentiated tumors [48].

The use of ^68^Ga DOTA-peptides PET in combination with ^18^F-FDG PET in GEP-NENs patients provides complementary information and could also be used for the whole-body characterization of tumor heterogeneity, which cannot be assessed with tumor biopsy alone. The combined PET/CT approach can identify the highest-grade and most aggressive lesions by localizing the most highly positive ^18^F-FDG and negative SSTRs. This whole-body evaluation of tumor glycolytic activity, together with SSTR expression, can help in classification of patients and selection of candidates or SSA therapy or PRRT, and can have a significant impact on therapeutic options. In addition, the combined PET/CT approach scan may detect changes that may occur in molecular heterogeneity of the disease over time (e.g., dedifferentiation of lesions), thereby enabling the timely adjustment of treatment plans [49].

According to 2020 ESMO Clinical Practice Guidelines for the diagnosis, treatment, and follow-up in GEP NENs, it is recommended that whole-body SSTR imaging should be part of the tumor staging, preoperative imaging, and restaging. ^68^Ga-SSTR PET/CT is recommended, but if it is not available, then SSTR SPECT can be used, although it is much less sensitive. MRI is preferable to CT to detect hepatic, pancreatic, brain, and bone lesions, whereas CT should be used to evaluate lung lesions. The use of ^18^F-FDG-PET is optional in GEP-NENs and should be used individually by balancing potential advantages with costs [7].

Although the definite role of combining ^68^Ga SSTR and ^18^F-FDG PET in NENs has not been established, several studies have shown favorable results by using both studies in the diagnosis, as well as prognosis and outcome prediction.

Kayani et al. [50] revealed in a retrospective study in 38 NETs patients that the sensitivity for the detection of tumors of ^68^Ga-DOTATATE PET/CT alone was 82%, and that of ^18^F-FDG PET/CT alone was 66%, while the sensitivity of combined dual PET/CT was more than 90%. The ^68^Ga-DOTATATE uptake was significantly higher than ^18^F-FDG in low-grade NENs (SUV 29 vs. 2.9, *p* < 0.001), whereas there was significantly higher ^18^F-FDG uptake in high-grade NENs over ^68^Ga-DOTATATE (SUV 11.7 vs. 4.4). A significant correlation with prominent tumor uptake of ^68^Ga-DOTATATE or ^18^F-FDG and tumor grade on histology was reported. In a systematic review of Muffatti et al. [51], they also showed similar results at initial diagnosis and restaging after treatment. The sensitivity of ^68^Ga DOTA-peptide PET and ^18^F-FDG PET was 93% and 58%, respectively, whereas the sensitivity of dual tracer PET was higher with the level of 94%.

The ^68^Ga-DOTA-peptide PET/CT is an advantage imaging modality for NENs and seems to surpass ^18^F-FDG for imaging well-differentiated NETs. However, functional imaging with dual PET tracer has the ability for more complete tumor evaluation, especially in intermediate- and high-grade tumors (Figure 3).

The study of Naswa N et al. suggested the combined using dual PET tracer because SSTR PET/CT overcomes the low sensitivity of ^18^F-FDG PET/CT, and ^18^F-FDG PET/CT overcomes the low specificity of SSTR PET/CT when the lesions do not concentrate ^68^Ga-DOTA-NOC. Moreover, the combined dual PET studies could determine the appropriate treatment regimen in NENs patients (octreotide therapy or conventional chemotherapy) [52].

The concept of combining the two PET tracers can be based on cell differentiation, and the SUVmax of both PET/CT studies in the context of Ki-67. Zhang et al. [53] revealed a negative correlation between the SUVmax of ^68^Ga-DOTATATE and Ki-67 (*r* = −0.415; *p* ≤ 0.001; in addition, a positive correlation between Ki-67 and SUVmax of ^18^F-FDG was reported (*r* = 0.683; *p* ≤ 0.001). They suggested that ^68^Ga-DOTATATE PET should be performed in patients with well-differentiated NETs, and ^18^F-FDG would be appropriate for patients with Ki-67 ≥ 10%.

A low SUVmax of ^68^Ga-DOTA-peptide PET is associated with poor prognosis for PFS and OS in NETs patients. In well-differentiated NETs, there is a better prognosis compared with other grades of NETs. The SUVmax of ^68^Ga-DOTA-peptide PET can be used as a prognosis predictor [54]. Lee et al. reported that a low SUVmax on SSTR PET/CT independently predicts early failure with SSA monotherapy in well-differentiated grade 1/2 GEP-NET. The SUVmax of SSTR PET predicted treatment failure with sensitivity of 39% and specificity of 98%. A SSTR SUVmax below 18.35 was associated with a shorter PFS. In multivariate analysis, a low SSTR SUVmax was the only predictor of early treatment failure with a hazard ratio (HR) of 6.85. Using SSTR PET/CT has a very high specificity in identified patients who do not receive the benefit from SSA treatment [55]. However, in high-grade NETs, it is also necessary to evaluate with ^68^Ga-DOTA-peptide PET prior to PRRT treatment, due to evidence that high-grade NETs patients were positive on both ^68^Ga-DOTA-peptide and ^18^F FDG PET. You et al. revealed that all patients with high-grade poorly differentiated NETs had positive SSTR PET/CT, and their findings were not different from FDG PET/CT in identifying >10 lesions or distant disease [56].

In the study of Cingarlini et al. [57], the SUVmax of ^68^Ga-DOTA-TOC was significantly higher than the SUVmax of ^18^F-FDG in grade 1/2 pancreatic NET; however, the SUVmax of ^18^F-FDG was higher in G2 than G1. Similarly, Chen et al. [58] reported that SUVs from ^18^F-FDG PET patients with G2/G3 NET were significantly higher than in patients with G1, which is lower than in the latter subgroup.

The study by Nilica et al. [59] revealed that the variation of SUVmax in FDG PET study between baseline and follow-up period was associated with prognosis. When SUVmax was constant or slightly changed, it was associated with a good prognosis. An increased SUVmax of an FDG value above 40% is associated with a poor prognosis.

The FDG PET assessment is also useful to predict treatment response to ^177^Lu-PRRT in NETs patients. Severi et al. suggested that no patients with FDG-PET-negative had disease progression at the first follow-up after ^177^Lu-PRRT. Grade 2 NET with FDG PET-positive (SUV cutoff >2.5) were often correlated with more aggressive tumors. PET-positive lesions with grade 2 NET (about 32%) did not respond to PRRT alone, so they may benefit from a more intensive treatment strategy, such as combined PRRT and chemotherapy [41].

The results of Panagoitidis revealed a change in the treatment plan in 84 NETs patients (80.8%). In 22 patients (21.1%), the decision was changed on the basis of ^18^F-FDG results; in 32 (30.8%), on both scan results; and in 50 (48.1%), on the ^68^Ga-DOTATATE results only. The OS declined rapidly with an increase in grade (*p* = 0.001) at approximately 91 months for G1, 59 months for G2, and 48 months for G3 [60].

FDG PET should be performed during follow-up, especially if there are signs of progression on other imaging modalities. Nilica et al. found that patients who were FDG-negative at baseline but FDG-positive during follow-up after treatment strongly correlate with an increased risk of progression. Then, patients might develop ^18^F-FDG-positive lesions during follow-up. These findings support the advantage of routinely repeating ^18^F-FDG PET in the long-term follow-up of NENs [59]. Binderup et al. revealed that, during the one-year follow-up, 14 patients died. Thirteen of the 57 (23%) FDG-PET-positive patients died compared to 1 in 41 (2%) FDG-PET-negative patients. A positive FDG-PET finding was correlated with significantly higher risk of death with HR of 10.3. The FDG-PET-positive cases had significantly lower PFS compared with the FDG-PET-negative cases with HR of 9.4. The only predictor of PFS was SUVmax of >3, with an HR of 8.4 [61].

Based on the current review study of Evangelista et al., they also suggest that combined ^68^Ga-DOTA peptides and ^18^F-FDG should be regarded as the initial imaging method for clinically suspicious NET or NET of unknown primary origin [62]. The retrospective study of Partelli et al. of 49 patients with a cytologically and/or histologically proven diagnosis of pancreatic NET (pNET) who performed both ^68^Ga-DOTANOC and ^18^F-FDG PET/CT on the same day reported that the dual-tracer PET/CT imaging approach resulted in a sensitivity of 100% for the diagnosis of pNET.

In spite of the fact that dual-tracer PET/CT revealed different tumor grading in the same lesions, it was not reported as having a significant impact on the treatment decision, except in NENs with Ki-67 > 10 [63].

Table 4 the role of SSTR PET and FDG PET in GEP-NENs.

## 2. Dual PET Imaging Grading System

### 2.1. The NETPET Score

In 2017, Chan et al. developed a scoring system known as the “NETPET score”, whereby results from the combined approach using ^68^Ga-DOTA-peptides and ^18^F-FDG PET in the workup of patients with metastatic NENs were incorporated into a 0–5 category scale (P0–P5) [47]. According to this system, P1 indicates purely SSTRs-avid disease without ^18^F-FDG activity in any lesion, P5 indicates the presence of significant ^18^F-FDG-positive and no SSTRs-positive lesion, P0 indicates a normal scan on both ^18^F-FDG and ^68^Ga-DOTA-peptide PET, and P2–P4 indicate patients with lesions aviding both tracers with different intensity (as the score increases from 2 to 4, ^18^F-FDG uptake increases and ^68^Ga-DOTA-peptide uptake decreases). They developed this PET-based scoring system by retrospectively reviewing 62 pathologically proven metastatic NET patients (mostly GEP NENs), who performed both ^18^F-FDG and ^68^Ga-DOTATATE PET/CT studies within 31 days. The authors visually compared uptake of the ^68^Ga-DOTATATE PET and FDG PET by using the SUVmax scale from 0 to 15 for ^68^Ga-DOTATATE and 0 to 7 for FDG PET as the values used for reporting in clinical practice. This study revealed that the OS was significantly correlated with the NETPET score (*p* = 0.0018). SSTR PET SUVmax might predict the response to PRRT, even though FDG-positive/SSTR-negative lesions may respond to PRRT, as well. This study suggested that subjects with NET-PET scores of P4 and P5 (significant FDG-positive/negative SSTR disease) might not obtain adequate tumor control from only PRRT and should receive systemic chemotherapy alternately, and this hypothesis should be researched in further studies.

In 2021, Chan et al. also revealed comparable results in a retrospective multicenter study (2011–2018) of 38 patients with bronchial NENs who had both ^18^F-FDG and ^68^Ga-DOTATATE PET within 60 days of each other. The NETPET scores and histology were significantly correlated with OS in univariate analyses (*p* = 0.003 and *p* = 0.01, respectively). In the multivariate analysis, only the NETPET score showed significance (*p* = 0.03). The NETPET scores were significantly correlated with histological grade (*p* = 0.006, chi-squared test). They concluded that NETPET score is a prognostic biomarker in bronchial NENs, as well as GEP NENs. Whilst it needs to be validated in prospective studies, it holds significant importance as a biomarker for a wide variety of NENs [64].

The NETPET scoring system is quite interesting because the results of both scans are summarized in a single parameter. Because it is a retrospective analysis, therefore, it is not possible to analyze the reasons leading to imaging with the dual tracer, type of treatment received, and the impact on the result obtained. Therefore, these findings require prospective validation. Although at present it might be too early to use NETPET score in routine clinical practice, the current studies have suggested several points for future research. Beyond confirmatory prospective studies, the benefit of dual PET imaging should be investigated in changing management decisions based on the high cost of imaging, and also of treatment [65].

### 2.2. Three-Scale Grading System

In 2020, Karfis et al. [66] developed a PET-based scoring scheme in a retrospective study of 85 metastatic GEP-NENs patients that included information from SSTR PET and FDG PET and was undertaken within a period of 3 months of each other. They grouped the study population into three different imaging groups (C1, all lesions were FDG negative/SSTR positive; C2, one or more lesions were FDG positive, while all of them were SSTR positive; and C3, one or more lesions were FDG positive lesions, while at least one of them was SSTR negative). They compared the prognostic value of this PET-based grading scheme with histological grade based on Ki-67. The histological grade showed no statistically significant difference in median PFS between G1 and G2 (*p* = 0.34) but was significant between G2 and G3 (*p* < 0.001), whereas the median PFS was significantly higher in C1 compared to C2 and in C2 compared to C3. They concluded that the combined ^68^Ga-DOTATATE/^18^F-FDG PET imaging stratification represents the accurate phenotype of GEP-NENs at any time of disease and has a high prognostic value when compared to histological grade classification. Its value as a prognostic imaging biomarker would be further corroborated within prospective study and multicenter testing to establish inter-rater reliability.

Although the new PET-based scoring system currently requires further validation with prospective studies including a large number of patients, the study also supports the use of the dual PET/CT imaging approach as a biomarker in prognostication and treatment selection, especially in patients with GEP-NETs.

## 3. Economic Benefits and Cost-Effectiveness

Currently, GEP-NENs represent a potentially important healthcare burden because they are usually diagnosed incidentally or when the patient has symptoms associated with hormone overproduction. However, GEP-NENs are heterogeneous tumors with non-specific symptoms, resulting in extensive healthcare use and delayed diagnosis affecting survival. In a global online survey of 1928 NENs patients, the average time reported from onset of the first symptom to diagnosis was 52 months, with nearly 30% of responders reporting waiting 5 years or more for an officially NENs diagnosis [67].

The previous study of Shen C. et al. [68] identified 9319 elderly patients diagnosed with NENs (mostly GEP-NENs) between 2003 and 2011, based on the Surveillance, Epidemiology and End Results (SEER) Medicare. They examined the patients’ conditions that may be related to NENs, resource use, and expense during the year prior to diagnosis. They reported that NENs patients tend to be diagnosed with various conditions, such as hypertension (64% vs. 53%), abdominal pain (22% vs. 8%), heart failure (12% vs. 8%), diarrhea (6% vs. 2%), peripheral edema (5% vs. 4%), and irritable bowel syndrome (1% vs. 0.5%), as compared to the non-cancerous control group. They also had much higher resource usage, including the average number of outpatient visits (22 vs. 17), percentage of ER visits (21% vs. 12%), and hospitalizations (28% vs. 17%). Similarly, NEN patients had significantly higher average total ($14,600 vs. $9500), outpatient ($6000 vs. $4200), and inpatient costs ($8600 vs. $5200). This study found that NEN patients require higher use of resources and costs. This information can inform the allocation of resources tailored to NENs needs.

Nowadays, it is accepted that PET/CT can provide value for money in the management of patients with NENs by using a more precise staging capability as compared to conventional imaging. This can result in cost reduction by preventing unnecessary or ineffective treatment and the associated side effects. PET/CT can predict outcomes and helps the clinician to identify patients who require more or less aggressive treatment, thereby improving outcomes. PET/CT reduces the likelihood of pretreatment progression, thus leading to more appropriate treatment, and it may increase the chance of disease cure. The previous study of Froelich MF et al. [69] analyzed the cost-effectiveness of ^68^Ga-DOTA-TATE PET/CT compared to ^111^In-pentetreotide SPECT/CT and to CT alone in NENs detection. In the base-case investigation, ^68^Ga-DOTA-TATE PET/CT ended up with a total cost of $88,000, while CT ended up with a total cost of approximately $89,000. SPECT/CT ended up with a total cost of approximately $90,000. Thus, CT and SPECT/CT were occupied by ^68^Ga-DOTA-TATE PET/CT. The use of PET/CT remains a cost-effective strategy. This is due to the reduced cost of treatment associated with timely detection. Additional costs of ^68^Ga-DOTA-TATE PET/CT compared to CT alone are reasonable in terms of treatment cost-saving and better outcome. The previous study of Schreiter et al. [70] revealed comparable results in 51 GEP-NENs patients who underwent ^68^Ga-DOTATOC PET/CT (*n* = 29) or ^111^In-DTPA-octreotide (*n* = 22). The results revealed ^68^Ga-DOTATOC PET/CT provided total costs (cost of equipment, personnel, and material) of approximately 550€, while the total cost of ^111^In-DTPA-octreotide is approximately 830€. They concluded that ^68^Ga-DOTATOC PET/CT was considerably cheaper than ^111^In-DTPA-octreotide in both material and personnel costs. In addition, using ^68^Ga-DOTATOC PET/CT may also have the potential to reduce the need for additional investigations, such as CT or MRI, thereby reducing the consequential costs.

Despite a large amount of works in the published literature in the NENs area, there is a lack of economic research data and specific comparative results relevant to ^18^F-FDG PET imaging in NENs.

## 4. Conclusions

Dual-tracer imaging with ^68^Ga-DOTA-peptide and ^18^F-FDG PET/CT appears to be a reasonable alternative to tissue sampling, because of the ability to reflect two different aspects of tumor biology, including SSTR expression and glucose metabolism. Therefore, imaging with dual tracers is recommended for well-differentiated GEP-NENs with Ki-67 ≥ 10%, providing information for treatment selection of SSA, PRRT, and chemotherapy. The combined dual-tracer PET is useful for precise clinical management and also shows a clear linear correlation between SUVmax and Ki-67 index.

With references to the heterogeneous expression and complementary findings to histopathology, we suggest that dual tracer PET should be permitted in the evaluation of patients at the time of diagnosis and during post-treatment follow-up in biological behavior assessment and prognosis. Although there is still very limited evidence to support the routine use of combined ^68^Ga-DOTA-peptide and ^18^F-FDG PET in the initial workup, dual-tracer PET might play a key role in the characterization of GEP-NENs patients, especially in patients affected with advanced tumors with a high disease burden. They should be performed both in the initial staging and during follow-up. The actual value of a combined PET approach is that it demonstrates the heterogeneity of GEP-NENs in both low-grade and advanced tumors. A single-core biopsy may not accurately evaluate the complexity and the heterogeneity of GEP-NENs, and, in some cases, it may underestimate the tumor grade. In addition, PET/CT is considered the most economical approach for diagnosis and treatment planning of GEP-NENs.

## Figures and Tables

**Figure 1 diagnostics-12-00280-f001:**
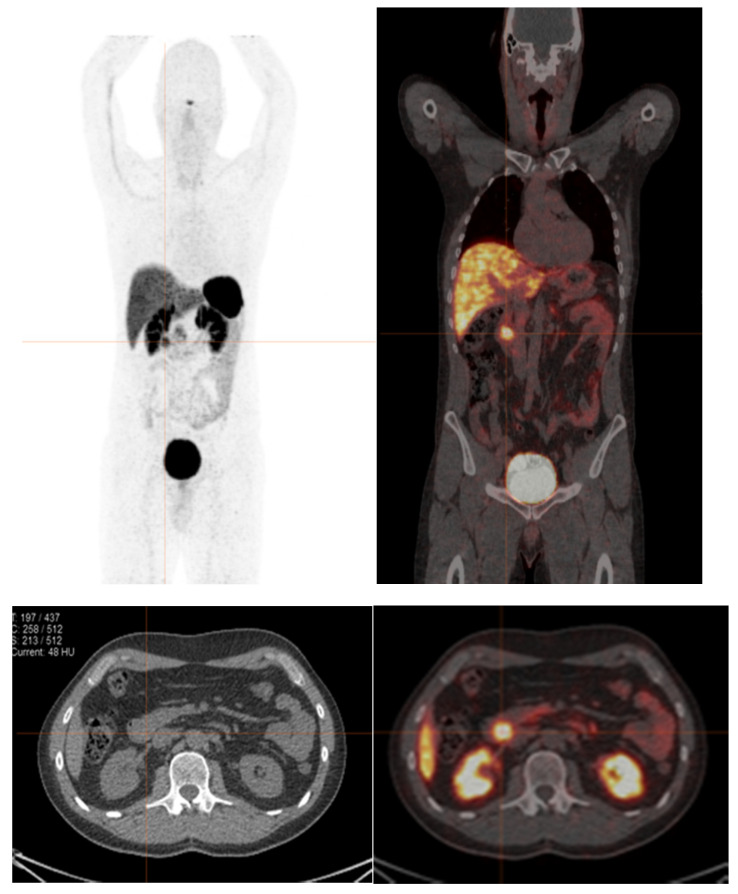
A 41-year-old male with recurrent NENs at the duodenum. ^68^Ga-DOTA-NOC PET/CT shows abnormal focal tracer uptake at the duodenum (SUVmax of 5.9), without other definite evidence of abnormal tracer uptake. Ki-67 index from endoscopic fine-needle aspiration of duodenum was 1%. It is possible that the FNA site may not fully reflect the true pathological grade of a patient with heterogeneity of cellular differentiation.

**Figure 2 diagnostics-12-00280-f002:**
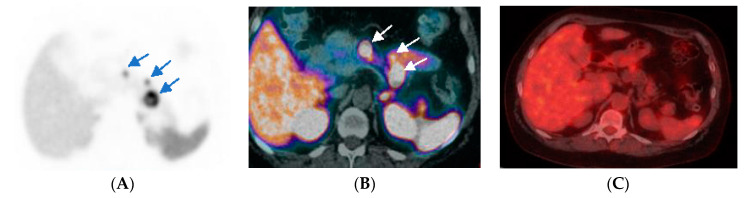
A patient with pathological proven pancreatic NENs (Ki-67 of 7%). ^68^Ga-DOTA-NOC PET image ((**A**), arrow) reveals 3 foci increased radiotracer uptake at the pancreas and also seen in axial ^68^Ga-DOTA-NOC PET/CT image ((**B**), arrow). However, the axial ^18^F-FDG PET/CT image (**C**) shows no abnormal FDG uptake in the pancreas at the same level.

**Figure 3 diagnostics-12-00280-f003:**
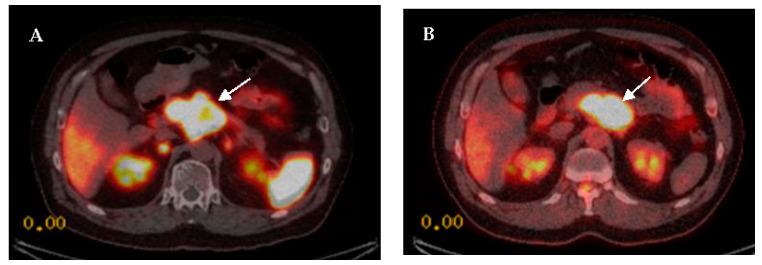
A patient with pathological proven pancreatic NENs (G3, Ki-67 of 30%). Axial ^68^Ga-DOTA-NOC PET/CT image ((**A**), arrow) reveals intense radiotracer uptake at the body of the pancreas and is also seen in axial ^18^F-FDG PET/CT image ((**B**), arrow). The value of combined PET studies is to demonstrate the heterogeneity of GEP-NENs, especially in high-grade NENs, and could determine the appropriate therapy.

**Table 1 diagnostics-12-00280-t001:** WHO 2019 classification for GEP-NENs [6].

Terminology	Differentiation	Grade	Mitotic Count (2 mm^2^) ^a^	Ki-67 Index (%)
NET, G1	Well differentiation	G1	<2	<3
NET, G2		G2	2–20	3–20
NET, G3		G3	>20	>20
NEC	Poorly differentiation	G3	>20	>20
	Small-cell			
	Large-cell			
MiNEN	Well or poorly differentiation	Variable	Variable	Variable

MiNEN, mixed neuroendocrine–non-neuroendocrine neoplasm; ^a^ 10 HPF = 2 mm^2^, at least 40 fields evaluated in areas of highest mitotic density.

**Table 2 diagnostics-12-00280-t002:** Clinical presentation of GEP-NENs by site of origin and by hormone presentation [7].

Type	Frequency	Symptoms	Secretory Product
Intestinal NENs	50% of GEP-NENs		
With carcinoid syndrome	20%	Flushing	Prostaglandin, Tachykinin, substance P
		Diarrhea	Serotonin
		Endocardial fibrosis	Serotonin
		Wheezing	Histamine, kinins, CgA
Without carcinoid syndrome	80%	Unspecific abdominal pain	CgA
Pan-NENs	30% of GEP-NENs		
Functioning	10–30%	Zollinger-Ellison syndrome	Gastrin
		Hypoglycemia	Insulin
		Necrolytic erythema, Hyperglycemia	Glucagon
		WDHA syndrome	VIP
		Diabetes, gallstones, diarrhea	Somatostatin
		Cushing syndrome	CRH, ACTH
		Acromegaly	GHRH, GH
		Hypercalcemia	PTHrP
		Flushing	Calcitonin
		Diarrhea	Serotonin, CgA
Non-functioning	70–90%	Unspecific abdominal pain	CgA
		Rarely jaundice (cholestasis and cholelithiasis)	Pancreatic polypeptide

ACTH, adrenocorticotropic hormone; CgA, chromogranin; CRH, corticotropin-releasing hormone; PTHrP, parathyroid-hormone-related peptide; VIP, vasoactive intestinal peptide; WDHA syndrome, watery diarrhea–hypokalemia–achlorhydria syndrome.

**Table 3 diagnostics-12-00280-t003:** Expression of somatostatin receptors in different GEP-NENs (%).

Tumor Types	Receptor Subtypes				
	SSTR 1	SSTR 2	SSTR 3	SSTR 4	SSTR 5
All GEP-NENs	68	86	46	93	57
Gastrinoma	33	50	17	83	50
Insulinoma	33	100	33	100	67
Glucagonoma	67	100	67	67	67
VIPoma	100	100	100	100	100
Mid-Gut NENs	80	95	65	35	75

**Table 4 diagnostics-12-00280-t004:** Role of SSTR PET and ^18^F FDG PET in GEP-NENs.

Role	Information
^68^Ga DOTA-peptide PET/CT	
Diagnosis (initial staging)	Localize primary tumor and detect metastatic site
	High sensitivity in the detection of well-differentiated and/or low-grade NENs (G1,G2)
Pretreatment evaluation	Determine SSTR status (positive SSTR patients are more likely to respond to targeted SSA treatment or PRRT)
Surveillance after treatment (re-staging)	Useful in monitoring patients with known disease in the detection of residual, recurrent, or progressive disease
Assessment of treatment response	Useful for monitoring response to therapy (surgery, radiotherapy, chemotherapy, and PRRT)
Prognosis	Well-differentiated NETs tend to have higher SUV on 68Ga-DOTA peptide PET with favorable prognosis, while lower SUV of 68Ga-DOTA-peptide PET may be associated with aggressive behavior and lead to poor prognosis
^18^F FDG PET/CT	
Diagnosis (initial staging)	High sensitivity in the detection of poorly differentiated and/or high-grade NETs (NEC) or G3 with negative SSTR imaging
Pretreatment evaluation	High FDG uptake is recommended for chemotherapy after SSA therapy or PRRT
	It is useful in delineating disease extent, particularly in aggressive and high-grade tumors
Surveillance after treatment/re-staging	Repeating ^18^F-FDG PET in the long-term follow-up, especially if signs of progression in other imaging methods are detected
	Patient with FDG-negative initially but FDG-positive during follow-up after treatment strongly correlates with a higher risk of progression
Assessment of treatment response	Useful for predicting response after PRRT; FDG-PET-negative after treatment tends to have a good response, but PET-positive were frequently associated with more aggressive disease and did not respond to PRRT; they may benefit from more intensive treatment, for example, the combined chemotherapy
Prognosis	A positive FDG-PET result was associated with significantly higher mortality risk and significantly lower PFS compared to the FDG-PET-negative result
ombined ^68^Ga DOTA-peptides and ^18^F FDG PET/CT	
Diagnosis (initial staging)	Should be considered in the patient with clinically suspected GEP-NENs or NET of unknown primary origin (regarding the heterogeneous expression and complementary findings to histopathology)
Pretreatment evaluation	Increased sensitivity in the detection of lesions as compared to only ^68^Ga-DOTA-peptide or FDG PET study alone
Assessment of treatment response	Combination of 2 studies can decide the mode of appropriate therapy (octreotide therapy or conventional chemotherapy)
Prognosis	Higher ^68^Ga-DOTA-peptide avidity and lower FDG avidity had favorable prognosis, while lower ^68^Ga-DOTA-peptide avidity and greater FDG avidity had poor prognosis

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
