# Peer review of "Role of Combined 68Ga DOTA-Peptides and 18F FDG PET/CT in the Evaluation of Gastroenteropancreatic Neuroendocrine Neoplasms"

_diagnostics, 2022, doi:10.3390/diagnostics12020280_

Round 1

Reviewer 1 Report

In this review article, the authors summarized the role of a combination of Ga-DOTA PET and FDG PET/CT for the management of GEP NET. It covered recent articles well and the conclusion is convincing. 

-    A native check is needed. There are many grammatical mistakes. 
-    The citation numbers are wrong ((35-36), (57-58))
-    In some countries, Ga-DOTA PET is not available so that In-111 pentetreotide SPECT/CT is still used for the management of NET. Thus, the authors may want to add a dedicated paragraph  (e.g., 1.2 or 1.3) that explains the diagnostic value of In-111 pentetreotide SPECT/CT
-    If the authors have some representative PET images, it would be better to provide the figure as this is a review article for the imaging. 

Author Response

Response to Reviewer 1 Comments

Point 1: A native check is needed. There are many grammatical mistakes.

Response 1: We thank the reviewer for this comment and already revised it. Our senior co-author “Professor Sobhan Vinjamuri” Head of nuclear medicine department, The Royal Liverpool University Hospital, who is a native-speaker and specialist in the article content, have proof read this manuscript before re-submitting.

Point 2: The citation numbers are wrong ((35-36), (57-58))

Response 2: We thank the reviewer for this comment and already revised it.

Point 3: In some countries, Ga-DOTA PET is not available so In-111 pentetreotide SPECT/CT is still used for the management of NET. Thus, the authors may want to add a dedicated paragraph  (e.g., 1.2 or 1.3) that explains the diagnostic value of In-111 pentetreotide SPECT/CT

Response 3: We thank the reviewer for this comment and already added it. Please see on page 4 (line 106-137) and page 5 (line 138-139, line 175-176), page 6 (line 177-196) as seen in red characters.

Point 4: If the authors have some representative PET images, it would be better to provide the figure as this is a review article for the imaging.

Response 4: We thank the reviewer for this comment and already added it as seen as Figures 1-3. Please see on page 6, 8 and 9

Please see the revised version of manuscript in the attached file

Reviewer 2 Report

The paper 'Role of combined 68Ga DOTA-peptides and 18F FDG PET / CT in the evaluation of gastroenteropancreatic neuroendocrine neoplasms', presented for review, summarizes the use of these compounds in supplementing cancer diagnostics. Work written carefully, with properly selected literature. The reviewer checked that there is in fact no literature item that would summarize the use of the presented compounds in cancer diagnostics. In addition, the subject is extremely interesting. The work encourages the reader to explore the subject and conduct further research. 

Author Response

We thank the reviewer for your opinion and valuable suggestions.

Please see the revised manuscript in the attached file

Round 2

Reviewer 1 Report

The quality of the manuscript has been improved significantly.